# Mystery of the Passerini Reaction for the Synthesis of the Antimicrobial Peptidomimetics against Nosocomial Pathogenic Bacteria

**DOI:** 10.3390/ijms25158330

**Published:** 2024-07-30

**Authors:** Deepak S. Wavhal, Dominik Koszelewski, Cezary Gulko, Paweł Kowalczyk, Anna Brodzka, Karol Kramkowski, Ryszard Ostaszewski

**Affiliations:** 1Institute of Organic Chemistry, Polish Academy of Sciences, Kasprzaka 44/52, 01-224 Warsaw, Poland; deepak.wavhal@icho.edu.pl (D.S.W.); dominik.koszelewski@icho.edu.pl (D.K.); cezary.gulko@icho.edu.pl (C.G.); anna.brodzka@icho.edu.pl (A.B.); 2Department of Animal Nutrition, The Kielanowski Institute of Animal Physiology and Nutrition, Polish Academy of Sciences, Instytucka 3, 05-110 Jabłonna, Poland; 3Department of Physical Chemistry, Medical University of Bialystok, Kilińskiego 1 Str., 15-089 Białystok, Poland; kkramk@wp.pl

**Keywords:** peptidomimetics, Passerini reaction, Gram-positive bacteria, Gram-negative bacteria, antimicrobial activity, nosocomial infections, MIC

## Abstract

The first example of applying salicylaldehyde derivatives, as well as coumarin with the formyl group at the C8 position in its structure, as carbonyl partners in a three-component Passerini reaction, is presented. As a result of research on the conditions of the Passerini reaction, the important role of the hydroxyl group in the salicylaldehyde used in the course of the multicomponent reaction was revealed. When an aldehyde with an unprotected hydroxyl group is used, only two-component α-hydroxy amide products are obtained. In contrast, the use of acylated aldehyde results in three-component α-acyloxy amide products with high efficiency. The developed protocol gives access to structurally diversified peptidomimetics with good yield. The compounds were also evaluated as antimicrobial agents against selected strains of nosocomial pathogenic bacteria. The structure–activity relationship revealed that inhibitory activity is strongly related to the presence of the trifluoromethyl group (CF_3_) or the methyl group at the C4 position in an unsaturated lactone ring of the coumarin scaffold. MIC and MBC studies were carried out on eight selected pathogenic bacteria strains (Gram-positive pathogenic *Staphylococcus aureus strain* (ATCC 23235), as well as on Gram-negative *E. coli* (K12 (ATCC 25404), R2 (ATCC 39544), R3 (ATCC 11775), and R4 (ATCC 39543)), *Acinetobacter baumannii* (ATCC 17978), *Pseudomonas aeruginosa* (ATCC 15442), and *Enterobacter cloacae* (ATCC 49141) have shown that the tested compounds show a strong bactericidal effect at low concentrations. Among all agents investigated, five exhibit higher antimicrobial activity than those observed for commonly used antibiotics. It should be noted that all the compounds tested showed very high activity against *S. aureus*, which is the main source of nosocomial infections that cause numerous fatalities. Additionally, the cytotoxicity of sixteen derivatives was measured with the use of the MTT test on BALB/c3T3 mouse fibroblast cell lines. The cytotoxicity studies revealed that the tested substances exert a similar or lower effect on cell proliferation than that observed for commonly used antibiotics within the range of therapeutic doses. A parallel MTT assay using ciprofloxacin, bleomycin, and cloxacillin showed that these antibiotics are more cytotoxic when tested in mammalian cells, and cell viability is in the range of 85.0–89.9%. Furthermore, we have shown that the studied coumarin-based peptidomimetics, depending on their structural characteristics, are nonselective and act efficiently against various Gram-positive and Gram-negative pathogens, which is of great importance for hospitalised patients.

## 1. Introduction

In today’s context, traditional strategies to combat infections caused by multidrug-resistant bacteria (MDR) are proving insufficient, compounded by the emergence of multidrug-resistant pathogens, posing an escalating global health threat. Consequently, there is an increasing demand within the pharmaceutical and medical sectors for innovative antibiotics. Antibiotic resistance has become a major global public health challenge [1,2]. The World Health Organisation (WHO) reports that in 2019, MDR directly caused an estimated 1.27 million deaths worldwide and played a role in 4.95 million deaths [3]. In 2023, a global WHO report warned of an imminent “post-antibiotic” epoch, where secondary bacterial infections for people undergoing surgeries, chemotherapy, or organ transplants could become untreatable [4,5,6].

Recent research highlights the importance of synthesising a diverse range of coumarin derivatives, which are known for their potential therapeutic value. Coumarins are widely recognised chemical compounds that are characterised by their simple structure, excellent solubility, favourable bioavailability, minimal toxicity, and other beneficial properties. They are commonly present in a variety of plant-based products and nutraceuticals. Due to their diverse biological and pharmacological effects, both naturally occurring and artificially synthesised coumarins have attracted considerable attention from researchers. These effects are associated with a variety of biological processes, including their ability to combat bacterial infections [7,8,9].

The strategic selection of molecular scaffolds in drug design plays a vital role in the search for novel therapeutic agents with improved efficacy and safety profiles. Among these scaffolds, coumarin, with its diverse pharmacological activities, has attracted considerable attention in medicinal chemistry. In particular, the exploration of various positions within the coumarin framework has led to the discovery of potent antimicrobial agents. However, despite extensive investigations of other positions, the C8 position of coumarin remains relatively underexplored in the context of antimicrobial activity [10]. Interestingly, preliminary studies have suggested the potential of the C8 position to increase antimicrobial potency, yet a comprehensive exploration of this moiety is conspicuously lacking in the literature. In particular, the diagram (Figure 1A) illustrates the pharmaceutical activity associated with each position of the coumarin molecule, with particular emphasis on the sparingly investigated C8 position [10]. Compound B, with a methyl substituent at the C8 position of coumarin, showed a three-fold decrease in IC_50_ compared to compound A without substitution, indicating a significant impact on antimicrobial activity [11] (Figure 1B). It should also be emphasised that antimicrobial substances of natural origin, Chlorobiocin and Novobiocin, have been identified, having in their structure a chlorine atom or a methyl group at the C8 position of the coumarin scaffold, respectively (Figure 1C). Motivated by this observation, we embarked on a systematic investigation to harness the unexplored antimicrobial potential of the C8 position through the design and synthesis of coumarin-based peptidomimetics. By targeting this position, we seek to elucidate the structural features that govern antimicrobial efficacy, paving the way for the development of novel therapeutics to combat infectious diseases. Multicomponent reactions have emerged as an efficient alternative to complex multistep organic synthesis methodologies for the synthesis of such compounds. Among these, the Passerini multicomponent reaction (MCR), named after the distinguished chemist Mario Passerini, stands out for its simplicity and adaptability. Using aldehydes, carboxylic acids, and isocyanides, the Passerini MCR enables the synthesis of α-acyloxyamide derivatives that are promising in antimicrobial applications. Following the Passerini reaction, the resulting product was identified as a peptidomimetic compound.

Peptidomimetics, as the name suggests, are small peptide-based compounds that mimic the physicochemical properties (e.g., structure, hydrophobicity) and biological activity (e.g., antimicrobial activity, mode of action) of antimicrobial proteins (AMPs). Peptidomimetics usually possess increased resistance to proteolytic degradation and a longer in vivo half-life due to the formation of unnatural backbone structures.

Many researchers have suggested that peptidomimetics are a solution to the antibiotic resistance crisis [12,13,14,15]. Due to their dual nature, they are believed to alter the integrity of bacterial cell membranes, causing cells to break down quickly and die [16,17,18,19]. Unlike many antibiotics, peptidomimetics do not need to enter bacterial cells to work, making them a powerful weapon against various antibiotic resistance mechanisms [20,21,22]. Furthermore, bacteria are less likely to develop resistance to peptidomimetics because these compounds target a part of the cell membrane that is similar across different bacterial species [23].

Our findings shed new light on the potential of combining the C8 position of coumarin with peptidomimetics as a viable approach to curbing antibiotic emergence (Figure 1D). This investigation aims to evaluate the bacterial responses to antimicrobial combinations in vitro and to explore the possible clinical implications of adjunct therapy with coumarin peptidomimetics.

## 2. Results

### 2.1. Chemistry

Peptidomimetics, due to their structural versatility and functional diversity, have a wide range of biological activities, including high solubility and significant efficacy against bacterial cell membranes, indicating their potential as powerful antimicrobial agents [24]. Peptidomimetics have been shown to have the ability to target and disrupt the integrity of bacterial cell membranes, causing compromised structural and functional integrity of bacterial cells [24].

In the initial phase of our investigation into the C8 position coumarin peptidomimetics, we selected salicylaldehyde as a model moiety to compare antimicrobial activity. Surprisingly, a review of the literature revealed that salicylaldehyde had not been used as a substrate in multicomponent reactions throughout the century-long history of the Passerini reaction. This intriguing observation led us to question why salicylaldehyde had been overlooked in such reactions. Consequently, we focused on the use of salicylaldehyde as a substrate in the Passerini reaction, which could also provide insight into the behaviour of 7-hydroxy-8-formyl-coumarin, whose structure is analogous to salicylaldehyde. Initially, we conducted the Passerini reaction in DCM using 4-bromo salicylaldehyde with phenylacetic acid and 4-methoxy benzyl isocyanide as substrates (Table 1). The choice of bromo-substituted salicylaldehyde was motivated by the potential for postreaction derivatization. In particular, we did not obtain the expected three-component Passerini product **B** (Figure 1); instead, a two-component product **A** was isolated (Figure 1). It should be noted that such compounds are typically challenging to synthesise and often require harsh conditions, such as TiCl_4_ as a catalyst [25].

Given these unexpected results, we explore the reaction using various solvents, as detailed in Table 1. Consistently, only the two-component product **A** was obtained across all solvents, with no product **B** formation in polar protic solvents such as ethanol. Subsequently, we repeated the reaction with unsubstituted salicylaldehyde, observing a reduced yield of the two-component product at a yield of 6%. Based on our previous research, we also used dimethyldioctadecylammonium bromide (DODAB) in water as the reaction medium, which similarly produced the two-component product (Table 1) [26].

In an attempt to resolve this ambiguity in the absence of the typical Passerini reaction product **B**, we protected the hydroxy group of salicylaldehyde with an acetyl group. Fortunately, this modification resulted in only the Passerini product **B** with an isolated yield in DCM as a solvent (Table 1).

To further explore the scope of the solvent and implement a green approach, we used our previously published H_2_O/DODAB protocol as the reaction medium. By varying the percentage of DODAB in water, we achieved product **B** with the highest isolated yield of 78% with 20% DODAB as an additive. Although the micellar system was shown to be effective in some reactions, the yield was significantly reduced when 4-fluorobenzyl isocyanide was used in this medium, compared to a yield of 70% in DCM as the solvent (Table 1).

The results underscored the importance of protecting the hydroxy group in facilitating the desired transformations and inspired our subsequent investigations with 7-hydroxycoumarin derivatives (Figure 2).

On the basis of our successful synthetic strategy using salicylaldehyde, we turned our attention to 7-hydroxycoumarin derivatives, namely 7-hydroxy coumarin (umbelliferone), 4-methyl-7-hydroxycoumarin, and 4-trifluoromethyl-7-hydroxycoumarin. To build the peptidomimetic scaffold at the C8 position, we formylated the coumarin at the C8 position using a well-known protocol of Duff aldehyde synthesis. Motivated by the reaction with salicylaldehyde, we tried a Passerini reaction without the protection of the hydroxy group, but the reaction did not proceed (Figure 2). Notably, we did not obtain the two-component product in the case of coumarin. After the protection of the hydroxy group with an acyl moiety, we successfully synthesised the Passerini product with coumarins. Unexpectedly, the acyl protection of the coumarin moiety was challenging because the acylated compound was not stable enough. In this instance, we performed the acylation of 7-hydroxy-8-formyl coumarin and used this compound directly for the next step without further purification (Figure 2).

We used three different isocyanides, which were benzyl isocyanide, 4-methoxy benzyl isocyanide, and 4-fluorobenzyl isocyanide, in the Passerini reaction, and phenylacetic acid was a constant component in all reactions (Figure 2). Target C8-formyl coumarin and salicylaldehyde-based peptidomimetics **1**–**16** were prepared with yields of up to 83%. The structures of the compounds obtained were confirmed using NMR, as provided in the Appendix A. Our synthetic strategy, inspired by the initial success with salicylaldehyde, demonstrates the versatility and efficacy of the Passerini reaction in the synthesis of peptidomimetics. Furthermore, our findings highlight the importance of hydroxy group protection in the resolution of synthetic challenges and in the achievement of desired product outcomes. These results not only expand the synthetic toolbox for the development of antimicrobial agents but also provide valuable insights into the strategic design and optimisation of peptidomimetic scaffolds for medicinal chemistry applications.

### 2.2. In Vitro Biological Studies of Synthesised Compounds

#### 2.2.1. MIC and MBC Studies

The effects of the 16 compounds investigated (illustrated in Figure 3) were evaluated within bacterial cells using a previously established methodology [9]. Following the determination of the minimum inhibitory concentration (MIC) and the minimum bactericidal concentration (MBC), the MIC values ranged between 0.25–4.5 µM, while the MBC values fell within the range of 1–8 (+/−0.5) µM for *E. coli* (K12 (ATCC 25404), R2 (ATCC 39544), R3 (ATCC 11775), and R4 (ATCC 39543)) and Gram-negative strains of *A. baumannii* (ATCC 17978), *P. aeruginosa* (ATCC 15442), *E. cloacae* (ATCC 49141) and *S. aureus strain* (ATCC 23235), as depicted in Figure 4. MIC serves as a reference for assessing the susceptibility or resistance of bacterial strains to the antibiotic applied in vitro. On the contrary, MBC represents the minimum concentration of an antibacterial agent required to eradicate bacteria, distinguishing it as bactericidal rather than bacteriostatic [27]. Basically, all our compounds tested against Gram-positive and Gram-negative bacterial strains show good activity against them. And all MIC values are below 5 µg/mL^−1^ (Figure 3). Interestingly, the change in the different isocyanides had a vital impact on activity. However, the coumarin ring plays a role in the activity against some bacteria, which we confirmed with the activity of compounds **1** and **15**. This means that the lactone ring in the coumarin had good potency compared to salicylaldehyde (Figure 2 and Figure 3). Interestingly, compounds **1**, **2**, **3**, **4**, and **9** show broad-spectrum activity against both Gram-negative and Gram-positive bacteria. Compound **9** shows very potent activity against the Gram-positive bacteria *S. aureus strain* (ATCC 23235).

Compounds **4** to **8** have greater efficacy against *A. aureus*, *P. aureus*, *Enterobacteria*, and *Staphylococcus* but showed moderate efficacy against *E. coli* stains. Compound **9** demonstrated a broad range and good action against all bacteria, as revealed confidentially with respect to the fluorine group. Compounds **7** and **8** also had a CF_3_ group, but their activity was less than that of compound **9**. Furthermore, compounds with a *para*-fluoro group in the isocyanide portion had less overall activity. Because umbelliferone exhibits poor antibacterial activity, the methyl and trifluoromethyl groups are essentially crucial to biological activity. Changes in isocyanides did not have a major effect on biological activity. Notably, in the salicylaldehyde peptidomimetics, comparing compounds **13**, **15**, and **16**, the addition of a bromine atom decreases the potency of molecules. The introduction of halogens into the chemical structure of peptides or peptoids is generally known to increase the hydrophobicity of molecules [28]. Therefore, halogenation, and particularly bromination, can be used to easily modify and alter the physicochemical and antibacterial properties of peptidomimetics, which remains consistent with reports in the literature on the modification of peptides [29].

Furthermore, as can be seen in Figure 3 and Figure 5, the investigated peptidomimetics generally exhibited greater antibacterial activity than the antibiotics commonly used (Appendix A). This is especially crucial because tested microorganisms that are resistant to known antibiotics have evidently increased, such as cloxacillin (clox) or ciprofloxacin (cipro) (Figure 5). The rate at which microorganisms will become resistant to bleomycin (bleo) is unknown, but this will cause a major disruption in the antibiotic arsenal that is now used to treat hospital infections. Interestingly, our compounds exhibit activity similar to that of bleomycin (bleo) (Appendix A) [9].

Antimicrobial agents are often classified as bacteriostatic or bactericidal. A drug is considered bactericidal if the MBC to MIC ratio is low (less than 4–6) and if it is feasible to achieve drug concentrations that eradicate 99.9% of the exposed organisms. Conversely, if the MBC to MIC ratio is high and it is not practical to achieve drug concentrations that kill 99.9% of bacteria, the drug is classified as bacteriostatic. The exact distinction between the bacteriostatic and bactericidal properties of many substances depends on the concentration of the pathogen and the drug that is reached in the target tissue. For all compounds tested **1**–**16**, we are dealing with bactericidal agents (Figure 6) [29].

#### 2.2.2. Oxidative DNA Damage Studies

Based on the results of the MIC and MBC toxicity tests in the bacterial model analysed), it was decided to modify the bacterial DNA with the analysed peptidomimetics and digest it with the Fpg enzyme (from the group of bifunctional repair glycosylases), which is a marker of oxidative stress (Figure 7 and Appendix A). Based on the results obtained from the modification of bacterial DNA after Fpg cleavage, we observed a very distinct change in the structure of the *E. coli* R4 (ATCC 39543) strain. This change was due to the ratios of the topological forms of the plasmid DNA, including ccc, linear, and oc forms. Additionally, we noted the formation of densely looped structures, known as concatemer, which may be caused by bacterial DNA topoisomerases (Appendix A). About 4% of the oxidative damage was identified after digestion with the Fpg protein, indicating that the analysed compounds damage the bacterial DNA very strongly due to oxidative stress induced by them within the cell (Figure 7) [27,28,29]. Our observations indicate that the formyl group at the C8 position in its structure as carbonyl partners of the analysed peptidomimetics may determine the toxicity of the *E. coli* R4 (ATCC 39543) strain, as evidenced by the MIC and MBC values. The results obtained were also statistically significant at the level of *p* < 0.05.

For further analysis with the Fpg protein (which is a bifunctional glycosylase that recognises oxidised DNA bases such as 8oxoG, FapyA, and FapyG) [9], we used plasmid DNA. Since we observed the highest number of modifications in the *E. coli* R4 (ATCC 39543) strain after treatment with the Fpg protein, we present these values as an exemplary model (Appendix A). The greatest damage, observed after modification with peptidomimetics and antibiotics after digestion with the Fpg protein, was visible in the *E. coli* R4 (ATCC 39543) strain, which is highlighted in the Appendix A. After treatment with the Fpg glycosade in the *E. coli* R4 (ATCC 39543) strain, we observed clearly visible damage in the forms of topological changes in the plasmid DNA forms: “ccc”, linear form, and ‘oc’. These forms were completely damaged and are visible only as stray bands (see Figure 8 and Appendix A), consistent with the data from the previous literature [9].

Model strains of *E.coli* (K12 (ATCC 25404), R2 (ATCC 39544), R3 (ATCC 11775), and R4 (ATCC 39543)), *S. aureus strain* (ATCC 23235), as well as *A. baumannii* (ATCC 17978), *P. aeruginosa* (ATCC 15442), and *E. cloacae* (ATCC 49141) strains, were statistically significance at *p* < 0.05 (Table 2) for all analysed compounds.

#### 2.2.3. Cytotoxicity Studies

Based on the MIC and MBC tests conducted in the bacterial strains analysed and after digestion with the Fpg protein altered with peptidomimetics and selected antibiotics, an additional MTT test was performed to determine the cytotoxicity of the compound using the BALB/c3T3 mouse embryonic fibroblast cell line (Figure 9). When using culture as a model for healthy cells under physiological conditions to obtain results highly correlated with those obtained in vivo, it is necessary to choose a cell line with genotypic and phenotypic characteristics as close as possible to those of normal cells. The choice of the appropriate culture is largely determined by its origin and purpose, according to the guidelines of the European Collection of Cell Cultures (ECACC) (Appendix A).

The BALB/c3T3 mouse embryonic fibroblast cells were treated with the tested peptidomimetics at concentrations ranging from 1 µg/mL to 8 µg/mL and were incubated for 24 h. All tested peptidomimetics were not cytotoxic to BALB/c3T3 cells at the lowest concentration tested of 1 µg/mL, with viability percentages remaining above 99.50%. However, a gradual reduction in viability was caused by tested compounds **1**–**16** at 2 µg/mL, with cell viability percentages ranging from 87.00% for compound **6** and 66.50% for peptidomimetic **15**. The highest concentrations (6 µg/mL and 8 µM) inhibited cell viability, which was lowered to 14.06% and 1.06%, respectively (Figure 9, Appendix A). The obtained results were used to calculate the half-maximal inhibitory concentration (IC_50_) after 24 h of incubation with the most active antimicrobial peptidomimetics **1**–**3**, **9, 14**, and **15**. The IC_50_ values for fibroblasts after 24 h of incubation ranged from 4.26 µg/mL for compound **1** to 7.22 µg/mL for peptidomimetic **14**.

Analogously to the tested compounds **1**–**16**, the MTT test was performed using representative antibiotics: ciprofloxacin (cipro), bleomycin (bleo), and cloxacillin (clox)) (Figure 10). Similar concentrations of these antibiotics were used in the studies. The obtained results indicate that the cytotoxicity of the tested peptidomimetics **1**–**16** is lower than or comparable to that observed for these widely used drugs.

## 3. Materials and Methods

### 3.1. Chemicals

The required reagents were purchased from Sigma-Aldrich, Poznan, Poland and utilised as received without further purification. Prior to use, the water and hexane mixtures were distilled, while other solvents of analytical grade were employed without additional drying or purification steps. Solvents and volatile reagents were subjected to evaporation under reduced pressure. Reactions were conducted in dry glass vessels under ambient conditions. For Thin-Layer Chromatography (TLC) analysis, Merck, Poznan, Poland silica gel plates 60 F_254_ were used. The crude mixtures, after solvent evaporation, were purified using column chromatography on Merck, Poznan, Poland silica gel 60/230–400 mesh, using appropriate hexane and ethyl acetate. The ^1^H and ^13^C nuclear magnetic resonance (NMR) spectra were recorded in Chloroform-d on Bruker 400 and Varian 500 MHz spectrometers, with TMS (Trimethylsilane) serving as an internal standard. Chemical shifts were reported in parts per million (ppm) and referenced to the residual deuterated solvent signal, while coupling constants (*J*) were recorded in Hertz (Hz). High-resolution mass spectra (HRMS) were acquired on the Maldi SYNAPT G2-S HDMS apparatus (Waters, Warsaw, Poland) equipped with a QqTOF analyser.

### 3.2. General Procedure (***1***) for the Synthesis of Coumarin α-Acyloxy Carboxamides ***1***–***9***

General procedure for the Passerini reactions: a mixture of the corresponding 8-formyl-4-methyl-2-oxo-2*H*-chromen-7-yl acetate, 8-formyl-2-oxo-2*H*-chromen-7-yl acetate or 8-formyl-2-oxo-4-(trifluoromethyl)-2*H*-chromen-7-yl acetate (1 mmol), phenylacetic acid (1 mmol), and corresponding isocyanides (1 mmol) was stirred in DCM at 300 rpm and room temperature for 18 h. After the completion of the reaction, the solvent was removed under vacuum. The resulting residue was purified using column chromatography (silica gel, eluent: ethyl acetate/hexanes, 6:4) to yield the target coumarin α-acyloxy carboxamides **1**–**16**. The yields of the derivatives are shown in Figure 2. The structures of the products were identified using their ^1^H and ^13^C NMR spectra in the electronic support information (ESI), and known compounds were compared with data from the literature, along with elemental analysis and high-resolution mass spectrometry (HRMS).

**1-(7-Acetoxy-4-methyl-2-oxo-2*H*-chromen-8-yl)-2-((4-fluorobenzyl)amino)-2-oxoethyl 2-phenylacetate (1):** Compound **1** was obtained according to the General method with a 62% yield (63 mg, 0.126 mmol) as an off-white solid; m.p.: 122–124 °C. ^1^H NMR (400 MHz, CDCl_3_) *δ* 7.61 (d*, J* = 8.7 Hz, 1H), 7.37–7.25 (m, 5H), 7.20–7.09 (m, 6H), 6.76 (s, 1H), 6.36 (d, *J* = 5.4 Hz, 1H), 6.26 (d, *J* = 1.3 Hz, 1H), 4.38 (d, *J* = 5.7 Hz, 2H), 3.60 (s, 2H), 2.41 (d, *J* = 1.2 Hz, 3H), 2.26 (s, 3H). ^13^C NMR (100 MHz, CDCl_3_) *δ* 174.8, 169.1, 168.8, 167.5, 159.2, 152.8, 151.98, 137.4, 133.7, 133.1, 129.3, 129.0, 128.7, 128.6, 128.5, 128.1, 127.5, 127.4, 127.1, 125.8, 119.2, 118.1, 116.9, 114.5, 66.0, 43.6, 41.2, 20.8, 18.8. HRMS calcd. for C_29_H_25_NO_7_Na [M+Na]^+^, 522.1526, found: 522.1529. Element. anal. for C_29_H_25_NO_7_ calc. C 69.73, H 5.04, N 2.80. found C 69.42, H 5.08, N 2.90.**1-(7-Acetoxy-4-methyl-2-oxo-2*H*-chromen-8-yl)-2-((4-methoxybenzyl)amino)-2-oxoethyl 2-phenylacetate (2):** Compound **2** was obtained according to the General method with a 70% yield (75 mg, 0.142 mmol) as an off-white solid; m.p.: 66–68 °C. ^1^H NMR (400 MHz, CDCl_3_) *δ* 7.61 (d, *J* = 8.6 Hz, 1H), 7.24–7.02 (m, 8H), 6.87 (d, *J* = 8.4 Hz, 2H), 6.74 (s, 1H), 6.33 (s, 1H), 6.26 (s, 1H), 4.32 (s, 2H), 3.83 (s, 3H), 3.60 (s, 2H), 2.41 (s, 3H), 2.26 (s, 3H). ^13^C NMR (100 MHz, CDCl_3_) *δ* 169.1, 168.8, 167.3, 161.7, 159.1, 152.9, 151.9, 151.8, 133.1, 129.6, 129.5, 129.4, 129.0, 129.0, 128.8, 128.7, 128.5, 127.3, 125.7, 121.3, 119.2, 118.1, 114.6, 114.0, 66.0, 55.3, 43.1, 41.2, 20.8, 18.8. HRMS calcd. for C_30_H_27_NO_8_Na [M+Na]^+^, 552.1637, found: 552.1634. Element. anal. for C_30_H_27_NO_8_ calc. C 68.05, H 5.14, N 2.65. found C 67.91, H 5.24, N 2.85.**1-(7-Acetoxy-2-oxo-2*H*-chromen-8-yl)-2-(benzylamino)-2-oxoethyl 2-phenylacetate (3):** Compound **3** was obtained according to the General method with a 72% yield (75 mg, 0.154 mmol) as an off-white solid; m.p.: 64–66 °C. ^1^H NMR (400 MHz, CDCl_3_) *δ* 7.65 (d, *J* = 9.6 Hz, 1H), 7.48 (d, *J* = 8.5 Hz, 1H), 7.39–7.26 (m, 5H), 7.21–7.12 (m, 5H), 7.08 (d, *J* = 8.5 Hz, 1H), 6.75 (s, 1H), 6.38 (d, *J* = 9.6 Hz, 2H), 4.43–4.34 (m, 2H), 3.61 (s, 2H), 2.25 (s, 3H). ^13^C NMR (100 MHz, CDCl_3_) *δ* 173.2, 169.0, 168.7, 167.3, 153.4, 152.1, 142.8, 137.4, 133.0, 129.0, 129.0, 128.8, 128.6, 128.1, 127.5, 127.4, 119.6, 116.9, 116.2, 100.2, 65.8, 43.6, 41.2, 20.8. HRMS calcd. for C_28_H_23_NO_7_Na [M+Na]^+^, 508.1371, found: 508.1372. Element. anal. for C_28_H_23_NO_7_ calc. C 69.27, H 4.78, N 2.89. found C 69.10, H 4.79, N 2.92.**1-(7-Acetoxy-2-oxo-2*H*-chromen-8-yl)-2-((4-methoxybenzyl)amino)-2-oxoethyl 2-phenylacetate (4):** Compound **4** was obtained according to the General method with a 63% yield (70 mg, 0.135 mmol) as an off-white solid; m.p.: 57–59 °C. ^1^H NMR (400 MHz, CDCl_3_) *δ* 7.64 (d, *J* = 9.6 Hz, 1H), 7.47 (d, *J* = 8.5 Hz, 1H), 7.24–7.10 (m, 7H), 7.07 (d, *J* = 8.5 Hz, 1H), 6.88 (d, *J* = 8.6 Hz, 2H), 6.73 (s, 1H), 6.38 (d, *J* = 9.6 Hz, 1H), 6.32 (s, 1H), 4.32 (qd, *J* = 14.4, 5.5 Hz, 2H), 3.83 (s, 3H), 3.60 (s, 2H), 2.25 (s, 3H). ^13^C NMR (100 MHz, CDCl_3_) *δ* 169.0, 168.7, 167.2, 159.1, 159.0, 153.4, 152.1, 142.8, 133.5, 133.0, 129.5, 129.4, 129.0, 128.9, 128.8, 128.6, 128.6, 127.4, 119.6, 117.0, 116.9, 116.1, 114.0, 109.7, 65.8, 55.3, 43.1, 41.2, 20.8. HRMS calcd. for C_29_H_25_NO_8_Na [M+Na]^+^, 538.1480, found: 538.1478. Element. anal. for C_29_H_25_NO_8_ calc. C 67.57, H 4.89, N 2.72. found C 67.78, H 5.10, N 2.96.**1-(7-Acetoxy-2-oxo-2*H*-chromen-8-yl)-2-((4-fluorobenzyl)amino)-2-oxoethyl 2-phenylacetate (5):** Compound **5** was obtained according to the General method with a 23% yield (50 mg, 0.099 mmol) as a pale yellow solid; m.p.: 61–63 °C. ^1^H NMR (400 MHz, CDCl_3_) *δ* 7.65 (d, *J* = 9.6 Hz, 1H), 7.48 (d, *J* = 8.5 Hz, 1H), 7.23–7.14 (m, 7H), 7.08 (d, *J* = 8.5 Hz, 1H), 7.04–7.00 (m, 2H), 6.73 (s, 1H), 6.38 (d, *J* = 9.6 Hz, 1H), 6.34 (d, *J* = 6.0 Hz, 1H), 4.40–4.25 (m, 2H), 3.62 (s, 2H), 2.28 (s, 3H). ^13^C NMR (100 MHz, CDCl_3_) *δ* 169.0, 168.7, 167.4, 163.5, 161.0, 159.0, 153.3, 152.1, 142.9, 133.2, 133.2, 133.1, 129.8, 129.7, 129.0, 129.0, 128.8, 127.4, 119.6, 116.9, 116.8, 116.1, 115.5, 115.3, 65.7, 42.9, 41.2, 20.8. HRMS calcd. for C_28_H_22_FNO_7_Na [M+Na]^+^, 526.1280, found: 526.1278. Element. anal. for C_28_H_22_FNO_7_ calc. C 66.80, H 4.40, N 2.78. found C 66.82, H 4.33, N 3.05.**1-(7-Acetoxy-4-methyl-2-oxo-2*H*-chromen-8-yl)-2-((4-fluorobenzyl)amino)-2-oxoethyl 2-phenylacetate (6):** Compound **6** was obtained according to the General method with a 63% yield (66 mg, 0.127 mmol) as an off-white solid; m.p.: 56–59 °C. ^1^H NMR (400 MHz, CDCl_3_) *δ* 7.62 (d, *J* = 8.7 Hz, 1H), 7.25–7.08 (m, 8H), 7.01 (t, *J* = 8.6 Hz, 2H), 6.74 (s, 1H), 6.35 (s, 1H), 6.26 (s, 1H), 4.33 (qd, *J* = 14.7, 5.7 Hz, 2H), 3.61 (s, 2H), 2.41 (s, 3H), 2.29 (s, 3H). ^13^C NMR (100 MHz, CDCl_3_) *δ* 169.1, 168.7, 167.5, 163.5, 161.0, 159.1, 152.8, 152.0, 151.9, 133.2, 133.2, 133.1, 129.8, 129.7, 129.0, 128.8, 127.4, 125.8, 119.2, 118.1, 116.8, 115.5, 115.3, 114.5, 65.9, 42.9, 41.2, 20.8, 18.8. HRMS calcd. for C_29_H_24_FNO_7_Na [M+Na]^+^, 540.1437, found: 540.1434. Element. anal. for C_29_H_24_FNO_7_ calc. C 67.31, H 4.67, N 2.71. found C 67.29, H 4.67, N 2.55.**1-(7-Acetoxy-2-oxo-4-(trifluoromethyl)-2*H*-chromen-8-yl)-2-(benzylamino)-2-oxoethyl 2-phenylacetate (7):** Compound 7 was obtained according to the General method with a 63% yield (66 mg, 0.127 mmol) as an off-white solid; m.p.: 54–57 °C. ^1^H NMR (400 MHz, CDCl_3_) *δ* 7.74 (dd, *J* = 8.9, 1.7 Hz, 1H), 7.40–7.28 (m, 3H), 7.26–7.07 (m, 8H), 6.76 (d, *J* = 3.0 Hz, 2H), 6.35 (t, *J* = 5.6 Hz, 1H), 4.38 (qd, *J* = 14.6, 5.7 Hz, 2H), 3.61 (d, *J* = 1.2 Hz, 2H), 2.26 (s, 3H). ^13^C NMR (100 MHz, CDCl_3_) *δ* 169.0, 168.4, 167.1, 157.2, 153.7, 152.9, 137.3, 132.9, 128.9, 128.9, 128.8, 128.8, 128.7, 128.7, 128.7, 128.1, 127.6, 127.5, 126.6, 126.6, 120.2, 117.8, 115.4, 115.3, 111.6, 65.7, 43.7, 41.2, 20.8. HRMS calcd. for C_29_H_22_F_3_NO_7_Na [M+Na]^+^, 576.1241, found: 576.1246. Element. anal. for C_29_H_22_F_3_NO_7_ calc. C 62.93, H 4.01, N 2.53. found C 62.81, H 4.12, N 2.62.**1-(7-Acetoxy-2-oxo-4-(trifluoromethyl)-2*H*-chromen-8-yl)-2-((4-fluorobenzyl)amino)-2-oxoethyl 2-phenylacetate (8):** Compound **8** was obtained according to the General method with a 63% yield (66 mg, 0.127 mmol) as a pale yellow solid; m.p.: 51–55 °C. ^1^H NMR (400 MHz, CDCl_3_) *δ* 7.74 (dd, *J* = 8.9, 1.6 Hz, 1H), 7.24–7.12 (m, 8H), 7.03 (t, *J* = 8.7 Hz, 2H), 6.76 (s, 1H), 6.74 (s, 1H), 6.30 (d, *J* = 5.7 Hz, 1H), 4.32 (ddd, *J* = 40.4, 14.7, 5.8 Hz, 2H), 3.61 (s, 2H), 2.30 (s, 3H). ^13^C NMR (100 MHz, CDCl_3_) *δ* 169.0, 168.4, 167.2, 163.5, 157.2, 153.7, 153.0, 133.1, 132.9, 129.8, 129.7, 129.2, 129.0, 128.8, 127.5, 126.6, 126.6, 122.6, 120.2, 117.6, 115.6, 115.4, 115.3, 111.6, 65.6, 42.9, 41.2, 20.8. HRMS calcd. for C_29_H_21_F_4_NO_7_Na [M+Na]^+^, 594.1153, found: 594.1152. Element. anal. for C_29_H_21_F_4_NO_7_ calc. C 60.95, H 3.70, N 2.45. found C 60.79, H 3.83, N 2.63.**1-(7-Acetoxy-2-oxo-4-(trifluoromethyl)-2*H*-chromen-8-yl)-2-((4-methoxybenzyl)amino)-2-oxoethyl 2-phenylacetate (9):** Compound **9** was obtained according to the General method with a 62% yield (60 mg, 0.107 mmol) as a pale yellow solid; m.p.: 54–58 °C. ^1^H NMR (400 MHz, CDCl_3_) *δ* 7.74 (d, *J* = 8.4 Hz, 1H), 7.22–7.09 (m, 8H), 6.88 (d, *J* = 8.7 Hz, 2H), 6.75 (d, *J* = 9.3 Hz, 2H), 6.30 (s, 1H), 4.31 (qd, *J* = 14.5, 5.7 Hz, 2H), 3.83 (s, 3H), 3.60 (s, 2H), 2.27 (s, 3H). ^13^C NMR (100 MHz, CDCl_3_) *δ* 169.0, 168.4, 167.0, 159.2, 157.3, 153.7, 152.9, 141.3, 132.9, 129.5, 129.4, 129.3, 129.0, 128.8, 127.5, 126.6, 126.6, 126.6, 126.5, 120.2, 117.8, 115.4, 115.3, 114.0, 111.6, 65.7, 55.3, 43.1, 41.2, 20.8. HRMS calcd. for C_30_H_24_F_3_NO_8_Na [M+Na]^+^, 606.1357, found: 606.1352. Element. anal. for C_30_H_24_F_3_NO_8_ calc. C 61.75, H 4.15, N 2.40. found C 61.52, H 4.17, N 2.54.

### 3.3. General Procedure (***2***) for the Synthesis of Coumarin α-Acyloxy Carboxamides ***10***–***16***

General procedure for the Passerini reactions: a mixture of the corresponding salicylaldehyde (1 mmol), phenylacetic acid (1 mmol), and the corresponding isocyanides (1 mmol) was stirred in water in the presence of DODAB at 300 rpm and room temperature for 18 h. After the completion of the reaction, the solvent was removed under vacuum. The resulting residue was purified using column chromatography (silica gel, eluent: ethyl acetate/hexanes, 6:4) to provide the target salicylaldehyde α-acyloxy carboxamides **10**–**16**. The yields of the derivatives are shown in Figure 2. The structures of the products were identified using their ^1^H and ^13^C NMR spectra provided in the electronic support information (ESI) and, for known compounds, were compared with data from the literature, along with elemental analysis and high-resolution mass spectrometry (HRMS).

**2-(4-Bromo-2-hydroxyphenyl)-2-hydroxy-*N*-(4-methoxybenzyl)acetamide (10):** Compound **10** was obtained according to the General method with an 8% yield (29 mg, 0.079 mmol) as an off-white solid; m.p.: 96–100 °C. ^1^H NMR (400 MHz, CDCl_3_) *δ* 12.12 (s, 1H), 8.49 (d, *J* = 8.8 Hz, 1H), 7.47 (t, *J* = 9.1 Hz, 1H), 7.21 (dd, *J* = 10.3, 8.6 Hz, 3H), 7.19–7.10 (m, 1H), 7.06 (dd, *J* = 8.8, 1.8 Hz, 1H), 6.96–6.69 (m, 3H), 4.50 (d, *J* = 5.9 Hz, 2H), 3.80 (s, 3H). ^13^C NMR (100 MHz, CDCl_3_) *δ* 188.5, 163.7, 161.5, 159.4, 134.6, 133.2, 129.3, 128.4, 123.2, 121.9, 114.3, 55.3, 43.3. Element. anal. for C_16_H_16_BrNO_4_ calc. C 52.48, H 4.40, N 3.82. found C 52.42, H 4.47, N 4.04.**2-Hydroxy-2-(2-hydroxyphenyl)-*N*-(4-methoxybenzyl)acetamide (11):** Compound **11** was obtained according to the General method with a 17% yield (20 mg, 0.069 mmol) as a pale yellow oil; ^1^H NMR (400 MHz, CDCl_3_) *δ* 11.84 (s, 1H), 8.60 (dd, *J* = 8.2, 1.7 Hz, 1H), 7.54 (ddd, *J* = 8.8, 7.3, 1.7 Hz, 1H), 7.38 (s, 1H), 7.28–7.26 (m, 1H), 7.25 (d, *J* = 2.1 Hz, 1H), 7.00 (dd, *J* = 8.5, 0.8 Hz, 1H), 6.91 (ddt, *J* = 9.5, 5.0, 2.0 Hz, 4H), 4.51 (d, *J* = 5.9 Hz, 2H), 3.80 (s, 3H). ^13^C NMR (100 MHz, CDCl_3_) *δ* 189.8, 163.7, 161.8, 159.4, 138.1, 133.7, 129.3, 128.7, 119.5, 118.6, 117.7, 114.3, 55.3, 43.2. Element. anal. for C_16_H_17_NO_4_ calc. C 66.89, H 5.96, N 4.88. found C 66.78, H 5.54, N 5.22.**1-(2-Acetoxy-4-bromophenyl)-2-((4-methoxybenzyl)amino)-2-oxoethyl 2-phenylacetate (12):** Compound **12** was obtained according to the General method with an 83% yield (90 mg, 0.170 mmol) as a pale yellow solid; m.p.: 107–108 °C. ^1^H NMR (400 MHz, CDCl_3_) *δ* 7.36 (dd, *J* = 8.3, 1.9 Hz, 1H), 7.30 (d, *J* = 1.9 Hz, 1H), 7.25–7.18 (m, 6H), 7.07 (d, *J* = 8.6 Hz, 2H), 6.88–6.81 (m, 2H), 6.22 (s, 1H), 6.02 (t, *J* = 5.0 Hz, 1H), 4.25 (ddd, *J* = 47.3, 14.5, 5.8 Hz, 2H), 3.81 (s, 3H), 3.64 (t, *J* = 9.3 Hz, 2H), 2.14 (s, 3H). ^13^C NMR (100 MHz, CDCl_3_) *δ* 169.2, 168.9, 166.9, 159.2, 151.6, 149.2, 136.9, 133.2, 130.9, 129.5, 129.2, 129.1, 128.8, 127.4, 127.0, 126.6, 123.4, 114.1, 70.2, 55.3, 42.9, 41.2, 20.7. Element. anal. for C_26_H_24_BrNO_6_ calc. C 59.33, H 4.60, N 2.66. found C 59.32, H 4.73, N 2.88.**1-(2-Acetoxy-4-bromophenyl)-2-(benzylamino)-2-oxoethyl 2-phenylacetate (13):** Compound **13** was obtained according to the General method with a 78% yield (72 mg, 0.160 mmol) as a white solid; m.p.: 122–124 °C. ^1^H NMR (400 MHz, CDCl_3_) *δ* 7.38–7.29 (m, 5H), 7.25–7.18 (m, 6H), 7.17–7.12 (m, 2H), 6.24 (s, 1H), 6.09 (s, 1H), 4.32 (ddd, *J* = 44.5, 14.7, 5.8 Hz, 2H), 3.67 (t, *J* = 9.3 Hz, 2H), 2.14 (s, 3H). ^13^C NMR (100 MHz, CDCl_3_) *δ* 169.3, 168.9, 167.0, 149.2, 137.4, 133.2, 130.9, 129.6, 129.1, 128.8, 128.7, 127.8, 127.7, 127.49, 127.0, 126.7, 123.4, 70.2, 43.4, 41.2, 20.7. Element. anal. for C_25_H_22_BrNO_5_ calc. C 60.50, H 4.47, N 2.82. found C 60.70, H 4.28, N 2.86.**1-(2-Acetoxyphenyl)-2-((4-methoxybenzyl)amino)-2-oxoethyl 2-phenylacetate (14):** Compound **14** was obtained according to the General method with a 73% yield (100 mg, 0.222 mmol) as a pale yellow oil. ^1^H NMR (400 MHz, CDCl_3_) *δ* 7.41–7.34 (m, 2H), 7.22 (dd, *J* = 9.0, 5.2 Hz, 6H), 7.09 (dd, *J* = 11.9, 8.4 Hz, 3H), 6.84 (d, *J* = 8.6 Hz, 2H), 6.29 (s, 1H), 6.18 (t, *J* = 5.5 Hz, 1H), 4.26 (ddd, *J* = 40.3, 14.5, 5.8 Hz, 2H), 3.79 (s, 3H), 3.65 (s, 2H), 2.15 (s, 3H). ^13^C NMR (100 MHz, CDCl_3_) *δ* 169.5, 169.4, 167.4, 159.1, 148.8, 133.4, 130.3, 129.9, 129.7, 129.2, 129.1, 128.7, 127.8, 127.3, 126.3, 123.2, 114.0, 70.7, 55.3, 42.8, 41.1, 20.8. Element. anal. for C_26_H_25_NO_6_ calc. C 69.79, H 5.63, N 3.13. found C 69.38, H 5.71, N 3.02.**1-(2-Acetoxyphenyl)-2-(benzylamino)-2-oxoethyl 2-phenylacetate (15):** Compound **15** was obtained according to the General method with a 65% yield (83 mg, 0.197 mmol) as a pale yellow oil; ^1^H NMR (400 MHz, CDCl_3_) *δ* 7.45–7.25 (m, 6H), 7.25 (d, *J* = 1.2 Hz, 2H), 7.20 (d, *J* = 8.0 Hz, 2H), 7.17–7.10 (m, 3H), 6.32 (s, 1H), 6.21 (t, *J* = 5.4 Hz, 1H), 4.33 (ddd, *J* = 36.2, 14.7, 5.9 Hz, 2H), 3.67 (s, 2H), 2.15 (s, 3H). ^13^C NMR (100 MHz, CDCl_3_) *δ* 169.5, 169.5, 167.5, 148.8, 137.6, 133.4, 130.4, 129.9, 129.2, 128.7, 128.7, 127.8, 127.6, 127.4, 126.4, 123.2, 70.7, 43.4, 41.1, 20.8. Element. anal. for C_25_H_23_NO_5_ calc. C 71.93, H 5.55, N 3.36. found C 71.52, H 5.73, N 3.14.**1-(2-acetoxy-4-bromophenyl)-2-((4-fluorobenzyl)amino)-2-oxoethyl-2-phenylacetate (16):** Compound **16** was obtained according to the General method with a 62% yield (60 mg, 0.107 mmol) as a pale yellow solid; m.p.: 122–124 °C; ^1^H NMR (400 MHz, CDCl_3_) δ 7.58 (s, 1H), 7.43–7.28 (m, 8H), 7.29–7.18 (m, 3H), 6.95 (dd, *J* = 9.1, 8.3 Hz, 2H), 6.25 (s, 1H), 3.75 (s, 2H), 2.24 (s, 3H). ^13^C NMR (100 MHz, CDCl_3_) δ 169.4, 169.2, 165.0, 160.7, 158.3, 149.1, 133.4, 131.2, 129.9, 129.0, 127.6, 126.7, 126.6, 123.8, 121.4, 121.3, 115.7, 115.5, 70.0, 41.2, 20.8. Element. anal. for C_25_H_21_BrNO_5_ calc. C 58.38, H 4.12, N 2.72. found C 58.25, H 4.08, N 2.63.

### 3.4. Microorganisms and Media

The reference bacterial strains of *E.coli* (K12 ATCC 25404, R2 ATCC 39544, R3 ATCC 11775, and R4 ATCC 39543) and *Staphylococcus aureus strain* (ATCC 23235), as well as *Acinetobacter baumannii* (ATCC 17978), *Pseudomonas aeruginosa* (ATCC 15442), and *Enterobacter cloacae* (ATCC 49141) were obtained from LGC Standards U.K. and were used according to the recommendation of ISO 11133 [30]. These strains were used to test the antibacterial activity of the analysed compounds by determining the minimum inhibitory concentration (MIC) and minimum bactericidal concentration (MBC), as described in [8,9,26,27,28,29,30,31]. The results are shown in Appendix A.

### 3.5. Minimum Inhibitory Concentration (MIC) and Minimum Bactericidal Concentration (MBC)

MIC and MBC were estimated using a microtiter plate method with sterile 48- or 96-well plates, which is precisely described in [9]. Briefly, MIC and MBC, defined as the lowest concentration of a bacteriostatic agent, were determined. Briefly, 50 μL of the analysed compounds and the appropriate bacterial strains were added to the first row of the plate. Then, 25 μL of sterile Tryptone Soya Broth (TSB) medium was added to the other wells, and serial dilutions were performed. Subsequently, 200 μL of inoculated TSB medium containing resazurin (0.02 mg/mL) as an indicator was added to all wells. TSB medium was inoculated with 10^6^ colony-forming units (CFU)/mL (approximately 0.5 McFarland units) of the bacterial strains. The plates were incubated at 30 °C for 24 h. Changes in colour from blue to pink or yellowish with turbidity were considered positive, and the lowest concentration at which no visible change in colour occurred was recorded as the MIC, according to Kowalczyk et al. [32]. Each experiment (both MIC and MBC) was repeated at least three times.

To estimate MBC, dehydrogenase activity was determined by measuring the visible changes in colour from triphenyl tetrazolium chloride (TTC) to triphenyl formazan (TF). First, 4 mM dense culture (approximately 10^7^ CFU/mL) incubated in TSB medium at 25 °C for 24 h was placed in identical test tubes. A stock solution of analogues was prepared in DMSO at a concentration of 1 mM. Both solvent and negative controls were used in the tests. A positive control was used in the tests (an antibacterial agent with a known MIC against bacteria). The latest CLSI recommendations suggest that if a carbapenemase is detected in a strain of enteric bacillus, MIC values should be determined for carbapenems, a group of β-lactam antibiotics related to penicillin and cephalosporins, but chemically distinct. The appropriate compounds were then added to the test tubes until the mixture reached a final concentration of 10 to 250 mg/mL. The cultures were then incubated at 30 °C for 1 h. The test tubes were then sealed with parafilm and incubated for 1 h at 30 °C in the dark. The lowest concentration at which no visible red colour (formazan) appeared was taken as the MBC.

### 3.6. MTT Assay

The cytotoxic effects of the tested peptidomimetics **1**–**16** on BALB/c3T3 mouse fibroblast cells were determined using the MTT assay after 24 h of incubation at five different concentrations (1, 2, 4, 6, and 8 µg/mL). The MTT test is based on the ability of the mitochondrial dehydrogenase enzymes to convert an orange, water-soluble tetrazolium salt (3-(4,5-dimethylthiazol-2-yl)-2,5-diphenyltetrazolium bromide) into an insoluble formazan, which is a dark blue product of the above reaction. After dissolving the formazan crystals in DMSO or isopropanol, a coloured solution was formed, the intensity of which is measured spectrophotometrically within the wavelength range of 492–570 nm. The amount of coloured reduced MTT is proportional to the oxidative activity of the cell’s mitochondria and, under strictly defined experimental conditions, to the number of metabolically active (living) cells in the population. The MTT test can also be used to determine cell viability in populations of cells that no longer divide but are metabolically active. The MTT test is currently the most commonly used to assess cytotoxic activity and is recommended as a reference by international standards-setting organisations [33,34].

### 3.7. Estimation of Oxidised Damage Based on Bacterial DNA Digestion by the Fpg Protein

Based on MIC values, the *E.coli* R4 (ATCC 39543)) strain was selected for further DNA analysis. The 16 compounds interfere with DNA structure even when digested with the Fpg protein (formamidopyrimidine DNA N-glycosylase/AP lyase) in vitro during a 24-h experiment. The digestion of the Fpg protein with plasmids isolated from both control and cultures treated with peptidomimetics, regardless of the peptidomimetic structure, clearly showed visible damage, including mutual changes in covalently closed circle (ccc), linear, and open circle (oc) forms, as well as in fuzzy bands known as the ‘smear’ (Appendix A). In the case of plasmids from the *E. coli* R4 (ATCC 39543) strain, three traditional forms were observed: very poor form oc, linear, and ccc. We observed significant differences between control and peptidomimetic-modified plasmids in the electrophoretic images (Appendix A). Briefly, bacterial DNA was isolated from 2 mL of fresh medium using a New England Biolabs Kit (Labjot, Warsaw, Poland) according to the manufacturer’s instructions. The isolated DNA was digested by Fpg protein after modification with each analysed compound at 1 mg/mL, incubated with the analysed peptidomimetic, including the formyl group at the C8 position in its structure as the carbonyl partner in a three-component Passerini reaction (New England Biolabs, cat no. M0240S, 8000 U/mL). The Fpg protein was diluted 50 times with 10× NEB buffer (provided by the Fpg protein manufacturer) and mixed with 100× BSA solution (also supplied with the Fpg protein). Next, 8 µL of purified bacterial DNA was mixed with 2 µL of Fpg solution and 2 µL of NE Buffer and incubated at 37 °C for 30 min. Control bacterial DNA (incubation without tested compounds) and digested and undigested genomic DNA (incubated with the analysed compounds) samples were evaluated using 1% agarose gel electrophoresis. The DNA concentration was determined spectrophotometrically using the A260/A280 ratio. The level of oxidative damage was estimated using Image Quant software TL 10.2.

### 3.8. Statistical Analysis

All experimental data from at least three different trials (n = 3) are given as means ± standard error of the mean (SEM, manufacturer, Saint Louis, MO, USA). To compare pairs of means, the Tukey post hoc test was used, indicating statistical significance with * *p* < 0.05, ** *p* < 0.1, and *** *p* < 0.01 [10].

## 4. Conclusions

In this study, we successfully explored the use of salicylaldehyde as a substrate in the Passerini reaction, addressing a century-old gap in the literature. Our investigation began with the use of 4-bromo salicylaldehyde, which did not produce the expected three-component product but instead resulted in a challenging-to-synthesise two-component product under mild conditions. This prompted further exploration with different solvents, consistently yielding the two-component product and highlighting the need for substrate modification. By protecting the hydroxy group of salicylaldehyde with an acetyl group, we achieved the desired three-component Passerini product with an isolated yield of 86%, thus providing a novel approach to utilising salicylaldehyde in multicomponent reactions. Our research also demonstrated the efficacy of the H_2_O/DODAB system, achieving notable yields, particularly with 20% DODAB in water. However, this system showed limitations with certain isocyanides, underscoring the necessity for solvent optimisation. The coumarin-peptidomimetics synthesised via the multicomponent reaction, i.e., the Passerini reaction, inhibited the growth of harmful bacterial strains that cause various diseases. The combination of the two pharmacophores, coumarin and peptidomimetics, boosted antibacterial activity. This allowed us to study the underrated and overlooked C8 position of coumarin, which may be a promising area for future research on coumarin-based antimicrobial medicines. The CF_3_ and Me groups in the coumarin scaffold demonstrated an improved effect on the antimicrobial activity against Gram-positive bacteria staphylococcus as well as Gram-negative bacteria including *A. baumannii* (ATCC 17978), *P. aeruginosa* (ATCC 15442), and *Enterobacter*, and various strains of *E. coli* (K12 (ATCC 25404), R2 (ATCC 39544), R3 (ATCC 11775), and R4 (ATCC 39543)). Among the coumarin peptidomimetics, compounds **1**–**4** and **9** demonstrated high selectivity and antimicrobial activity comparable to or superior to antibiotics such as ciprofloxacin, bleomycin, and cloxacillin. The MTT assay on the BALB/c3T3 mouse fibroblast cell line revealed that the investigated peptidomimetics when used at therapeutic concentration dosages, affect the viability of selected eukaryotic cells similarly to commonly used antibiotics. Our research indicates that these peptidomimetics possess high antimicrobial efficacy with broad-spectrum activity against various pathogens, which is crucial for patient care and combating nosocomial bacteria in hospital settings. It is important to note that a low MBC/MIC ratio indicates the high toxicity of the tested compounds against the pathogenic *Staphylococcus*. Consequently, the desired biocidal effect is achieved at very low concentrations of these compounds. The use of these agents induces significant oxidative stress and inhibits the replication machinery, which is critical for combating pathogenic strains. The quest for new, non-toxic compounds for human health based on coumarin derivatives that are lethal to bacterial cells will play a crucial role in antibacterial therapy during emerging pandemics of our time. Furthermore, overcoming antibiotic resistance by targeting new, environmentally adapted bacterial strains, known as super-resistant bacteria, could mark a significant breakthrough in neutralising and reducing the cytotoxicity of these pathogenic species.

## Data Availability

This is available at the request of those interested.

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
