# Peer review of "Mystery of the Passerini Reaction for the Synthesis of the Antimicrobial Peptidomimetics against Nosocomial Pathogenic Bacteria"

_ijms, 2024, doi:10.3390/ijms25158330_

Round 1

Reviewer 1 Report

Comments and Suggestions for Authors
1- Related to action mechanism:
The text refers to the active site of an enzyme, the induction of oxidative stress and inhibition of the replicative machinery. A mechanism of action of this type of antimicrobial is not clearly described and referenced, nor what enzyme they inhibit, etc. The possible biological mechanism of action must be better described.
2-Related to terminology: the term cytotoxicity is used indistinctly for antibacterial activity and cytotoxicity (virulence) of the pathogenic specie. I think that cytotoxicity should be only related to the mammalian cellular toxicity of a compound, experiments that are not conducted in this paper. 
3-Releted to reaction mechanism: 
Why is  favored the two-component product when OH is free, without TiCl4? It is known that this two-component reaction requires water, where does it come from in the case of reaction in organic solvents? Why does acetylating OH favor the Passerini reaction? An explanation must be given as to the reaction mechanism regarding these experimental facts.
-the acronym DODAB is not named
- reference 22 is missing the title!

Author Response

RESPONSE TO REVIEWERS

Dear Editor,

please find enclosed for your consideration the revised article ijms-3085424 entitled  “Mystery of the Passerini Reaction for the Synthesis of the Antimicrobial Peptidomimetics Against Multidrug-Resistant Bacteria.” by Kowalczyk et al.

               Firstly, we would like to express our gratitude to Reviewers for their suggestions that allowed us to considerably improve our manuscript. We have revised the text according to the suggestions and we hope that you will now find it suitable for publication in the Materials. Below, please find the detailed information on the changes in the manuscript with answers to all comments. All changes made in the manuscript were highlighted using tracking mode in Word Editor.

Reviewer 1

Reviewer 1.- 1- Related to action mechanism: The text refers to the active site of an enzyme, the induction of oxidative stress and inhibition of the replicative machinery. A mechanism of action of this type of antimicrobial is not clearly described and referenced, nor what enzyme they inhibit, etc. The possible biological mechanism of action must be better described.

Response: We are grateful to the reviewer for drawing attention to this aspect. We have made changes in the interpretation of our results related to the conversion of a halogen to the peptidomimetic structure. A modified interpretation of the results obtained has been incorporated into the revised manuscript. According to the reviewer's suggestion, the work was supplemented with research results obtained for the fpg protein responsible for processes related to oxidative stress. The research results and their interpretation were included in the manuscript and supporting materials. According to the Reviewer's suggestion, in the part regarding antimicrobial activity studies, we changed “cytotoxicity” to “antimicrobial activity”.

Reviewer 1.-2-Related to terminology: the term cytotoxicity is used indistinctly for antibacterial activity and cytotoxicity (virulence) of the pathogenic specie. I think that cytotoxicity should be only related to the mammalian cellular toxicity of a compound, experiments that are not conducted in this paper.

Response: We appreciate your attention to this aspect. In the title of the manuscript itself, we included the research goal based on the study of antimicrobial activity. Cytotoxicity concerns a broad aspect of activity; in our case, the emphasis was on compounds that inhibit selected bacterial tiers. We conducted additional cytotoxicity studies in which we determined the effects on mammalian cells based on MTT essay. The results of these studies and their description were included in the revised version of the manuscript. The obtained MTT essay results for the tested peptidomimetics compounds were also compared for standardly used antibiotics. The obtained results showed that the developed and obtained peptidomimetics have lower cytotoxicity (99.5% viability) than those used as a control antibiotic.

Reviewer 1.- 3-Releted to reaction mechanism: Why is  favored the two-component product when OH is free, without TiCl4? It is known that this two-component reaction requires water, where does it come from in the case of reaction in organic solvents? Why does acetylating OH favor the Passerini reaction? An explanation must be given as to the reaction mechanism regarding these experimental facts.

Response: We appreciate your attention. In case of using native salicylaldehyde reaction proceeds via cyclic intermediate which in the presence of water molecule is transferred in to final hydroxy amide what stays in agreement with the work of Yamada et al., Eur. J. Org. Chem. 2015, 296–301 (Scheme 1). The content of water present in none-dry solvent used is sufficient for the described reaction to proceed. In case of O-acylated salicylaldehyde the reaction proceeds via classical known mechanism involving an imidate intermediate which undergoes Mumm rearrangement to afford the Passerini product (Scheme 2) (Li, Jie Jack (2021), Li, Jie Jack (ed.), "Passerini Reaction", Name Reactions: A Collection of Detailed Mechanisms and Synthetic Applications, Cham: Springer International Publishing, pp. 424–426, doi:10.1007/978-3-030-50865-4_115, ISBN 978-3-030-50865-4, retrieved 24 October 2022).  

Scheme 1. Plausible mechanism of the reaction with salicylaldehyde leading to alpha-hydroxy amides

In case of O-acylated salicylaldehyde the reaction proceeds via classical known mechanism involving an imidate intermediate which undergoes Mumm rearrangement to afford the Passerini product (Scheme 2) (Li, Jie Jack (2021), Li, Jie Jack (ed.), "Passerini Reaction", Name Reactions: A Collection of Detailed Mechanisms and Synthetic Applications, Cham: Springer International Publishing, pp. 424–426, doi:10.1007/978-3-030-50865-4_115, ISBN 978-3-030-50865-4, retrieved 24 October 2022).

Scheme 2. Mechanism of Passerini reaction.

Reviewer 1.-the acronym DODAB is not named

Response: We are very grateful for your attention. The full name DODAB was included in the manuscript

Reviewer 1.- reference 22 is missing the title!

Response: We are very grateful for your attention. The title has been added to the reference literature mentioned

Reviewer 2 Report

Comments and Suggestions for Authors

The manuscript by Wavhal et al. reports the synthesis and biological evaluation of antimicrobial peptidomimetics based on the coumarin scaffold. In this work, the authors initially identified that the modification of the C8 position of coumarin has not been explored previously. They then synthesized a library of C8-substituted coumarin-based analogues via the Passerini reaction. The minimum inhibitory concentrations (MICs) and minimum bactericidal concentrations (MBCs) of the synthesized analogues were determined, and the authors eventually identified compounds 1-4 and 9 as the best compounds in this study.

While the authors managed to optimize the yield of the Passerini reaction, the overall scope of the manuscript is very limited. The synthesis involved in this manuscript is very simple and straightforward while only MIC and MBC assays were carried out in this study. There is no information provided on the antibacterial activity of the analogues against virulence factors (e.g. inhibiting biofilm formation, disrupting established biofilm) of bacteria, nor the mechanism of action of the synthesized analogues. The cytotoxicity of the analogues against mammalian cells was also not determined. Hence, I would not recommend the publication of this manuscript in IJMS.

Detailed comments are as below:

1. There is only a very short paragraph in the introduction on peptidomimetics. The authors should expand this paragraph to include more background information of peptidomimetics.

2. Figure 1C and 1D was not explained / described in the manuscript.

3. Lines 113 and 115, peptidomimetics showed high antibacterial activity by disrupting the cell membrane of bacteria, not cell wall.

4. Line 188, sub-section title 2.2 should be “Antibacterial studies of synthesised compounds”

5. In lines 219-221, the authors claimed that the bulky bromine atom could hinder the ability of the analogues to fit into the active site of enzyme. What is the mechanism of action of the synthesized analogues, and what enzyme are the authors referring to?

6. In Figure 3, the authors should specify that strains K12, R2, R3, and R4 are E. coli K12, E. coli R2, E. coli R3 and E. coli R4 respectively. The x-axis label is also missing in Figure 3.

7. The authors claimed that the synthesized analogues act against multidrug-resistant bacteria, but the authors did not make it clear if any of the strain used is multidrug-resistant strain.

8. The paragraph in lines 242-250 is repeated in lines 255-263.

9. Some essential information is missing from the methodology section of the manuscript. Although some methodology has been previously described and the corresponding references have been provided in the manuscript, the author should still mention the methodology briefly (including essential information of the methods) in the manuscript.

For example, what was the concentration of the bacteria used in the MIC and MBC assays? Was the initial stock of analogues prepared in DMSO or water? If DMSO was used, what was the final concentration of DMSO in the assays? Was a solvent control and/or negative control used in the assays? Was a positive control (antibacterial agent with known MIC against the bacteria) used in the assays? If not, why not?

10. The authors should determine the cytotoxicity of the analogues against mammalian cells.

11. There are some typos and formatting errors in the manuscript. The authors should read the manuscript carefully and fix all the mistakes.

Author Response

RESPONSE TO REVIEWERS

Dear Editor,

please find enclosed for your consideration the revised article ijms-3085424 entitled  “Mystery of the Passerini Reaction for the Synthesis of the Antimicrobial Peptidomimetics Against Multidrug-Resistant Bacteria.” by Kowalczyk et al.

               Firstly, we would like to express our gratitude to Reviewers for their suggestions that allowed us to considerably improve our manuscript. We have revised the text according to the suggestions and we hope that you will now find it suitable for publication in the Materials. Below, please find the detailed information on the changes in the manuscript with answers to all comments. All changes made in the manuscript were highlighted using tracking mode in Word Editor.

Reviewer 2

Reviewer 2. While the authors managed to optimize the yield of the Passerini reaction, the overall scope of the manuscript is very limited. The synthesis involved in this manuscript is very simple and straightforward while only MIC and MBC assays were carried out in this study.

Response: We are grateful for this note. The aim of our research was to check the antimicrobial activity of derivatives of coumarin substituted at the C8 position and peptidomimetics based on the structure of salicylaldehyde. It should be noted that the desired products of the three-component Passerini reaction could not be obtained using procedures known in the literature. Only the conducted research demonstrated the key role of the hydroxyl group on the course of the Passerini reaction, and the selection of an appropriate protecting group made it possible to obtain target peptidomimetics.

Detailed comments are as below:

Reviewer 2.1. There is only a very short paragraph in the introduction on peptidomimetics. The authors should expand this paragraph to include more background information of peptidomimetics.

Response: According to Reviewer suggestion introduction was supplemented with additional and essential  information regarding peptidomimetics

Reviewer 2.2. Figure 1C and 1D was not explained / described in the manuscript.

Response: According to Reviewer suggestion Figure 1c and 1d was described.   

Reviewer 2.-3. Lines 113 and 115, peptidomimetics showed high antibacterial activity by disrupting the cell membrane of bacteria, not cell wall.

Response: We are very grateful. According to the Reviewer's suggestion, the title of the subsection was changed

Reviewer 2.-4. Line 188, sub-section title 2.2 should be “Antibacterial studies of synthesised compounds”

Response: We are very grateful. According to the Reviewer's suggestion, the title of the subsection was changed

Reviewer 2.-. In lines 219-221, the authors claimed that the bulky bromine atom could hinder the ability of the analogues to fit into the active site of enzyme. What is the mechanism of action of the synthesized analogues, and what enzyme are the authors referring to?

Response: We are grateful to the reviewer for drawing attention to this aspect. We have made changes in the interpretation of our results related to the conversion of a halogen to the peptidomimetic structure. A modified interpretation of the results obtained has been incorporated into the revised manuscript.

Reviewer 2.-. 6. In Figure 3, the authors should specify that strains K12, R2, R3, and R4 are E. coli K12, E. coli R2, E. coli R3 and E. coli R4 respectively. The x-axis label is also missing in Figure 3.

Response: We are grateful. According to the Reviewer suggestion E. coli were specified. The axis description is placed below the figures.

Reviewer 2.7. The authors claimed that the synthesized analogues act against multidrug-resistant bacteria, but the authors did not make it clear if any of the strain used is multidrug-resistant strain.

Response: We are grateful to the reviewer for drawing attention to this aspect. Our antimicrobial activity results were compared with those for routinely used antibiotics. It should be noted that the response of the bacterial strains selected by us, which belong to the pathogenic group, to the antibiotics used is highly varied and, in the case of ciprofloxacin and cloxacillin, requires the use of high doses, which may indicate that these strains are gradually becoming resistant to ciprofloxacin and cloxacillin.

Reviewer 2.8. The paragraph in lines 242-250 is repeated in lines 255-263.

Response: Mentioned paragraph was revised and corrected.

Reviewer 2. 9. Some essential information is missing from the methodology section of the manuscript. Although some methodology has been previously described and the corresponding references have been provided in the manuscript, the author should still mention the methodology briefly (including essential information of the methods) in the manuscript.

For example, what was the concentration of the bacteria used in the MIC and MBC assays? Was the initial stock of analogues prepared in DMSO or water? If DMSO was used, what was the final concentration of DMSO in the assays? Was a solvent control and/or negative control used in the assays? Was a positive control (antibacterial agent with known MIC against the bacteria) used in the assays? If not, why not?

Response: We are grateful to the reviewer for drawing attention to this aspect. According to the Reviewer suggestion methodology part of the manuscript was supplemented with additional information regarding protocols used.

Reviewer 2.10. The authors should determine the cytotoxicity of the analogues against mammalian cells.

Response: We are grateful to the reviewer for drawing attention to this aspect. According to Reviewer suggestion cytotoxicity was determined and the obtained results with discussion was provided to the manuscript.

The MTT test was conducted to determine the cytotoxicity of the compound using the BALB/c3T3 mouse embryonic fibroblast cell line. The choice of this type of cell as a model in the culture depends on the nature of the planned experiments. Cell banks, e.g.the European Collection of Cell Cultures (ECACC), have material from various organs and representing different properties.In studies in which the culture is to be a model of healthy cells, occurring in physiological conditions, in order to obtain results highly correlated with those obtained in in vivo conditions, it is necessary to choose the one whose geno- and phenotypic features are as close as possible to the features of normal cells, while in the case of testing the activity of cytostatic drugs, the tests are first performed in cancer cell cultures.The choice of the appropriate culture is largely determined by its origin.

Reviewer 2.11. There are some typos and formatting errors in the manuscript. The authors should read the manuscript carefully and fix all the mistakes.

Response: The manuscript was revised again and we made every effort to correct any linguistic or typographical errors

Round 2

Reviewer 2 Report

Comments and Suggestions for Authors

The authors have addressed most of the issues raised by the reviewer and amended the manuscript accordingly. However, the following aspects still need to be addressed before the manuscript can be considered for publication:

1. The newly added sentences regarding peptidomimetics are out of place. These sentences should either stand alone as a new paragraph or should be merged into the next paragraph on peptidomimetics.

2. In Figures 3-5, the authors still have not specified in the labels that strains K12, R2, R3, and R4 are E. coli K12, E. coli R2, E. coli R3 and E. coli R4 respectively.

3. As additional biological results and discussion have been added to the manuscript, the authors should divide section 2.2 into three sub-sections. For example:

a. “2.2 Antibacterial studies of synthesised compounds” should be changed to “2.2 In vitro biological studies of synthesised compounds”

b. After the section title, a sub-section title (e.g. “2.2.1 MIC and MBC studies”) should be added.

c. Before lines 267 and 309, a new sub-section title should be added.

4. While the authors claimed that the synthesized analogues act against multidrug-resistant bacteria, the authors still have not clarified or mentioned explicitly whether the bacterial strains used are multidrug resistant strain. It should be noted that compounds showing high antibacterial activity (e.g. low MIC and MBC) against a non-resistant bacterial strain do not necessarily show antibacterial activity against the corresponding multidrug-resistant strain. Compounds need to be tested against the actual multidrug-resistant bacterial strain before a conclusion can be drawn on whether these compounds can kill and/or inhibit the growth of multidrug-resistant bacteria.

5. Axis labels are missing in Figures 7-10.

6. In the experimental, the authors mentioned that the MTT assay was carried out at 1 μM of the synthesised compounds. However, such concentration is below the MIC and/or MBC of most of the compounds. It is unclear if the compounds are toxic to mammalian cells at the therapeutic dosage. The authors should determine the IC50 value of the compounds against mammalian cells or the mammalian cell viability at MBC or above the MBC of the compounds.

7. There are linguistic mistakes in the manuscript, especially in the newly added paragraphs and sentences. Moreover, there are still many typos and formatting errors in the manuscript:

- The authors should check that all compound numbers are in bold.

- The authors should check that all microorganism names are in italic.

- Line 77: “50” in “IC50” should be in subscript.

- Line 83: “position C8 position” should be “C8 position”.

- Line 133: “analogue” should be “analogous”.

- Line 169: “synthesis” should be “synthetic”.

- Line 211: “-1” should be in superscript.

- Line 212: “was playing” should be “plays”.

- Line 226: The sentence “Notably, our compounds were proved to be the fact” is confusing and seems incomplete.

- Experimental: The authors should check that all position descriptors (e.g. tert, N, H) are in italic.

- Line 541: “6” in “106 colony-forming units” should be in superscript.

- Line 550: “1 mM/ml” is incorrect.

- Line 566: The word “line” should be deleted.

Only some examples are listed above. The authors should read the manuscript carefully and fix all other mistakes.

Comments on the Quality of English Language

There are many typos, stylistic and linguistic mistakes throughout the manuscript. The authors need to read the manuscript carefully to fix all mistakes.

Author Response

RESPONSE TO REVIEWERS

Dear Editor,

please find enclosed for your consideration the revised article ijms-3085424 entitled  “Mystery of the Passerini Reaction for the Synthesis of the Antimicrobial Peptidomimetics Against Multidrug-Resistant Bacteria.” by Kowalczyk et al.

               Firstly, we would like to express our gratitude to Reviewers for their suggestions that allowed us to considerably improve our manuscript. We have revised the text according to the suggestions and we hope that you will now find it suitable for publication in the Materials. Below, please find the detailed information on the changes in the manuscript with answers to all comments. All changes made in the manuscript were highlighted using tracking mode in Word Editor.

Response: Due to the reviewer’s suggestion regarding antimicrobial studies on nosocomial pathogens but not on multidrug resistant bacterial strains, we have modified the title of the manuscript. Modified title is “Mystery of the Passerini Reaction for the Synthesis of the Antimicrobial Peptidomimetics Against Nosocomial Pathogenic Bacteria”. In our opinion, the new title fits to the matter of our studies specified in the submitted manuscript.

Reviewer: The authors have addressed most of the issues raised by the reviewer and amended the manuscript accordingly. However, the following aspects still need to be addressed before the manuscript can be considered for publication:

Response: We are very grateful for this conclusions and further remarks which improve quality of our manuscript

Reviewer: 1. The newly added sentences regarding peptidomimetics are out of place. These sentences should either stand alone as a new paragraph or should be merged into the next paragraph on peptidomimetics.

Response: We are very grateful for this remark. Mentioned sentence was placed as a new paragraph.

Reviewer:  2. In Figures 3-5, the authors still have not specified in the labels that strains K12, R2, R3, and R4 are E. coli K12, E. coli R2, E. coli R3 and E. coli R4 respectively.

Response: We are very grateful for this remark. According to Reviewer’s suggestion mentioned Figures were revised and modified.

Reviewer: 3. As additional biological results and discussion have been added to the manuscript, the authors should divide section 2.2 into three sub-sections. For example:

Response: We are very grateful for this remark. According to Reviewer’s suggestion additional sub-section were added.

Reviewer:  a. “2.2 Antibacterial studies of synthesised compounds” should be changed to “2.2 In vitro biological studies of synthesised compounds”

Response: We are very grateful for this remark. According to Reviewer’s suggestion additional sub-section were added.

Reviewer: b. After the section title, a sub-section title (e.g. “2.2.1 MIC and MBC studies”) should be added.

Response: We are very grateful for this remark. According to Reviewer’s suggestion additional sub-section were added.

Reviewer: c. Before lines 267 and 309, a new sub-section title should be added.

Response: We are very grateful for this remark. According to Reviewer’s suggestion additional sub-section were added.

Reviewer: 4. While the authors claimed that the synthesized analogues act against multidrug-resistant bacteria, the authors still have not clarified or mentioned explicitly whether the bacterial strains used are multidrug resistant strain. It should be noted that compounds showing high antibacterial activity (e.g. low MIC and MBC) against a non-resistant bacterial strain do not necessarily show antibacterial activity against the corresponding multidrug-resistant strain. Compounds need to be tested against the actual multidrug-resistant bacterial strain before a conclusion can be drawn on whether these compounds can kill and/or inhibit the growth of multidrug-resistant bacteria.

Response:  We are very grateful for this remark. We completely agree with the Reviewer's comment. Our research was conducted on pathogens present in hospitalized patients. Therefore, we changed the title of the manuscript to remove drug-resistant pathogens and reformulated the conclusions to better reflect the nature of the results obtained.

Reviewer: 5. Axis labels are missing in Figures 7-10.

Response:  We are very grateful for this remark. According to Reviewer’s suggestion Axis labels were added.

Reviewer: 6. In the experimental, the authors mentioned that the MTT assay was carried out at 1 μM of the synthesised compounds. However, such concentration is below the MIC and/or MBC of most of the compounds. It is unclear if the compounds are toxic to mammalian cells at the therapeutic dosage. The authors should determine the IC50 value of the compounds against mammalian cells or the mammalian cell viability at MBC or above the MBC of the compounds.

Response:  We are very grateful for this remark. As suggested by the Reviewer, we repeated the MTT tests using different concentrations, also within the range of therapeutic concentrations resulting from the MBC tests. We also calculated IC50 values ​​for the most active peptidomimetics. The modified figures were included in the revised manuscript. We did similar research for antibiotics for comparative purposes.

Reviewer: 7. There are linguistic mistakes in the manuscript, especially in the newly added paragraphs and sentences. Moreover, there are still many typos and formatting errors in the manuscript:

Response:  We are very grateful for this remark. The manuscript was revised and corrected.

Reviewer: - The authors should check that all compound numbers are in bold.

Response:  We are very grateful for this remark. According to Reviewer’s suggestion all compound numbers were changed to bold.

Reviewer: The authors should check that all microorganism names are in italic.

Response:  We are very grateful for this remark. According to Reviewer’s suggestion microorganism names were verified and changed to italic.

Reviewer: - Line 77: “50” in “IC50” should be in subscript.

Response:  We are very grateful for this remark. According to Reviewer’s suggestion this was modified

Reviewer:- Line 83: “position C8 position” should be “C8 position”.

Response:  We are very grateful for this remark. According to Reviewer’s suggestion this was modified

Reviewer: Line 133: “analogue” should be “analogous”.

Response: We are very grateful for this remark. According to Reviewer’s suggestion this was corrected

Reviewer: - Line 169: “synthesis” should be “synthetic”.

Response: We are very grateful for this remark. According to Reviewer’s suggestion this was corrected

Reviewer: - - Line 211: “-1” should be in superscript.

Response:  We are very grateful for this remark. According to Reviewer’s suggestion this was corrected

Reviewer:- Line 212: “was playing” should be “plays”.

Response:  We are very grateful for this remark. According to Reviewer’s suggestion this was corrected

Reviewer:- Line 226: The sentence “Notably, our compounds were proved to be the fact” is confusing and seems incomplete.

Response:  We are very grateful for this remark. According to Reviewer’s suggestion mentioned sentence was revised and removed.

Reviewer:-- Experimental: The authors should check that all position descriptors (e.g. tert, N, H) are in italic.

Response:  We are very grateful for this remark. According to Reviewer’s suggestion this was corrected

Reviewer:-- Line 541: “6” in “106 colony-forming units” should be in superscript.

Response:  We are very grateful for this remark. According to Reviewer’s suggestion this was corrected

Reviewer:-- Line 550: “1 mM/ml” is incorrect.

Response:  We are very grateful for this remark. According to Reviewer’s suggestion this was corrected

Reviewer:-- Line 566: The word “line” should be deleted.

 Response:  We are very grateful for this remark. According to Reviewer’s suggestion this was deleted.

Round 3

Reviewer 2 Report

Comments and Suggestions for Authors

The authors have significantly improved the manuscript compared to the original submission. I would recommend the publication of the manuscript after the following minor errors have been fixed:

1. Lines 219, 221, 222, 225, 227, 228, 233: “no.” should be deleted.

2. Unit should be added to the y-axis label in Figures 3-5.

3. Line 503: “107.108” should be “107-108”.

4. As the authors repeated the MTT assay with different concentrations of compound during the revision, line 582 (in the experimental section) needs to be updated to include the newly tested concentrations.

Author Response

Firstly, we would like to express our gratitude to Reviewer for their suggestions that allowed us to considerably improve our manuscript. We have revised the text according to the suggestions and we hope that you will now find it suitable for publication in the IJMS. The detailed changes made in the manuscript were in light blue colour.

Once again, we would like to thank you for your insightful suggestions which improved the scientific quality of our manuscript.

The authors have significantly improved the manuscript compared to the original submission. I would recommend the publication of the manuscript after the following minor errors have been fixed:

  1. Lines 219, 221, 222, 225, 227, 228, 233: “no.” should be deleted.

the word "no." was removed at the reviewer's suggestion

  1. Unit should be added to the y-axis label in Figures 3-5.

Units on the y-axis have been added

  1. Line 503: “107.108” should be “107-108”.

It has been corrected

  1. As the authors repeated the MTT assay with different concentrations of compound during the revision, line 582 (in the experimental section) needs to be updated to include the newly tested concentrations.

It has been corrected
